# On the Complexity and Approximation of Binary Evidence in Lifted Inference

**Guy Van den Broeck** and **Adnan Darwiche**
Computer Science Department
University of California, Los Angeles
{guyvdb,darwiche}@cs.ucla.edu

## Abstract

Lifted inference algorithms exploit symmetries in probabilistic models to speed up inference. They show impressive performance when calculating unconditional probabilities in relational models, but often resort to non-lifted inference when computing conditional probabilities. The reason is that conditioning on evidence breaks many of the model's symmetries, which can preempt standard lifting techniques. Recent theoretical results show, for example, that conditioning on evidence which corresponds to binary relations is #P-hard, suggesting that no lifting is to be expected in the worst case. In this paper, we balance this negative result by identifying the *Boolean rank* of the evidence as a key parameter for characterizing the complexity of conditioning in lifted inference. In particular, we show that conditioning on binary evidence with bounded Boolean rank is efficient. This opens up the possibility of approximating evidence by a *low-rank Boolean matrix factorization,* which we investigate both theoretically and empirically.

## 1 Introduction

Statistical relational models are capable of representing both *probabilistic* dependencies and *relational* structure [1, 2]. Due to their first-order expressivity, they concisely represent probability distributions over a large number of propositional random variables, causing inference in these models to quickly become intractable. Lifted inference algorithms [3] attempt to overcome this problem by exploiting *symmetries* found in the relational structure of the model.

In the absence of evidence, *exact* lifted inference algorithms can work well. For large classes of statistical relational models [4], they perform inference that is polynomial in the number of objects in the model [5], and are therein exponentially faster than classical inference algorithms. When conditioning a query on a set of evidence literals, however, these lifted algorithms lose their advantage over classical ones. The intuitive reason is that *evidence breaks the symmetries* in the model. The technical reason is that these algorithms perform an operation called shattering, which ends up reducing the first-order model to a propositional one. This issue is implicitly reflected in the experiment sections of exact lifted inference papers. Most report on experiments without evidence. Examples include publications on FOVE [3, 6, 7] and WFOMC [8, 5]. Others found ways to efficiently deal with evidence on only *unary* predicates. They perform experiments without evidence on binary or higher-arity relations. There are examples for FOVE [9, 10], WFOMC [11], PTP [12] and CP [13].

This evidence problem has largely been ignored in the exact lifted inference literature, until recently, when Bui et al. [10] and Van den Broeck and Davis [11] showed that conditioning on *unary* evidence is tractable. More precisely, conditioning on unary evidence is polynomial in the size of evidence. This type of evidence expresses attributes of objects in the world, but not relations between them. Unfortunately, Van den Broeck and Davis [11] also showed that this tractability does not extend to

evidence on *binary* relations, for which conditioning on evidence is #P-hard. Even if conditioning is hard in general, its complexity should depend on properties of the specific relation that is conditioned on. It is clear that some binary evidence is easy to condition on, even if it talks about a large number of objects, for example when all atoms are true ($\forall X, Y \ \mathrm{p}(X, Y)$) or false ($\forall X, Y \ \neg \mathrm{p}(X, Y)$). As our *first main contribution*, we formalize this intuition and characterize the complexity of conditioning more precisely in terms of the *Boolean rank* of the evidence. We show that it is a measure of how much lifting is possible, and that one can efficiently condition on large amounts of evidence, provided that its Boolean rank is bounded.

Despite the limitations, useful applications of exact lifted inference were found by sidestepping the evidence problem. For example, in lifted generative learning [14], the most challenging task is to compute partition functions without evidence. Regardless, the lack of symmetries in real applications is often cited as a reason for rejecting the idea of lifted inference entirely (informally called the "death sentence for lifted inference"). This problem has been avoided for too long, and as lifted inference gains maturity, solving it becomes paramount. As our *second main contribution*, we present a first general solution to the evidence problem. We propose to *approximate evidence* by an over-symmetric matrix, and will show that this can be achieved by minimizing Boolean rank. The need for approximating evidence is new and specific to lifted inference: in (undirected) probabilistic graphical models, more evidence typically makes inference easier. Practically, we will show that existing tools from the data mining community can be used for this low-rank Boolean matrix factorization task.

The evidence problem is less pronounced in the *approximate* lifted inference literature. These algorithms often introduce approximations that lead to symmetries in their computation, even when there are no symmetries in the model. Also for approximate methods, however, the benefits of lifting will decrease with the amount of symmetry-breaking evidence (e.g., Kersting et al. [15]). We will show experimentally that over-symmetric evidence approximation is also a viable technique for approximate lifted inference.

## 2 Encoding Binary Relations in Unary

Our analysis of conditioning is based on a reduction, turning evidence on a binary relation into evidence on several unary predicates. We first introduce some necessary background.

### 2.1 Background

An *atom* $\mathrm{p}(t_1, \ldots, t_n)$ consists of a predicate $\mathrm{p}/n$ of arity $n$ followed by $n$ arguments, which are either (lowercase) *constants* or (uppercase) *logical variables*. A *literal* is an atom $a$ or its negation $\neg a$. A formula combines atoms with logical connectives (e.g., $\vee, \wedge, \Leftrightarrow$). A formula is *ground* if it does not contain any logical variables. A *possible world* assigns a truth value to each ground atom. *Statistical relational languages* define a probability distribution over possible words, where ground atoms are individual random variables. Numerous languages have been proposed in recent years, and our analysis will apply to many, including MLNs [16], parfactors [3] and WFOMC problems [8].

**Example 1.** The following MLNs model the dependencies between web pages. A first, peer-to-peer model says that student web pages are more likely to link to other student pages.

$$w \quad \mathrm{studentpage}(X) \wedge \mathrm{linkto}(X, Y) \Rightarrow \mathrm{studentpage}(Y)$$

It increases the probability of a world by a factor $e^w$ with every pair of pages $X, Y$ that satisfies the formula. A second, hierarchical model says that professors are more likely to link to course pages.

$$w \quad \mathrm{profpage}(X) \wedge \mathrm{linkto}(X, Y) \Rightarrow \mathrm{coursepage}(Y)$$

In this context, *evidence* $e$ is a truth-value assignment to a set of ground atoms, and is often represented as a conjunction of literals. In *unary evidence*, atoms have one argument (e.g., $\mathrm{studentpage}(a)$) while in *binary evidence*, they have two (e.g., $\mathrm{linkto}(a, b)$). Without loss of generality, we assume full evidence on certain predicates (i.e., all their ground atoms are in $e$).[1] We will sometimes represent unary evidence as a Boolean vector and binary evidence as a Boolean matrix.

**Example 2.** Evidence $e = \mathrm{p}(a, a) \wedge \mathrm{p}(a, b) \wedge \neg\, \mathrm{p}(a, c) \wedge \cdots \wedge \neg\, \mathrm{p}(d, c) \wedge \mathrm{p}(d, d)$ is represented by

$$\mathbf{P} = \begin{array}{c} \\ X{=}a \\ X{=}b \\ X{=}c \\ X{=}d \end{array} \begin{array}{cccc} \mathrm{p}(X,Y) & Y{=}a & Y{=}b & Y{=}c & Y{=}d \\ \left[\begin{array}{cccc} 1 & 1 & 0 & 0 \\ 1 & 1 & 0 & 1 \\ 0 & 0 & 1 & 0 \\ 1 & 0 & 0 & 1 \end{array}\right] \end{array}$$

We will look at computing *conditional probabilities* $\Pr(q \mid e)$ for single ground atoms $q$. Finally, we assume a representation language that can express universally quantified logical constraints.

## 2.2 Vector-Product Binary Evidence

Certain binary relations can be represented by a pair of unary predicates. By adding the formula

$$\forall X,\ \forall Y,\ \mathrm{p}(X, Y) \Leftrightarrow \mathrm{q}(X) \wedge \mathrm{r}(Y) \tag{1}$$

to our statistical relational model and conditioning on the q and r relations, we can condition on certain types of binary p relations. Assuming that we condition on the q and r predicates, adding this formula (as hard clauses) to the model does not change the probability distribution over the atoms in the original model. It is merely an indirect way of conditioning on the p relation.

If we now represent these unary relations by vectors $\mathbf{q}$ and $\mathbf{r}$, and the binary relation by the binary matrix $\mathbf{P}$, the above technique allows us to condition on any relation $\mathbf{P}$ that can be factorized in the outer vector product $\mathbf{P} = \mathbf{q}\,\mathbf{r}^{\mathsf{T}}$.

**Example 3.** Consider the following outer vector factorization of the Boolean matrix $\mathbf{P}$.

$$\mathbf{P} = \begin{bmatrix} 0 & 0 & 0 & 0 \\ 1 & 0 & 0 & 1 \\ 0 & 0 & 0 & 0 \\ 1 & 0 & 0 & 1 \end{bmatrix} = \begin{bmatrix} 0 \\ 1 \\ 0 \\ 1 \end{bmatrix} \begin{bmatrix} 1 \\ 0 \\ 0 \\ 1 \end{bmatrix}^{\mathsf{T}}$$

In a model containing Formula 1, this factorization indicates that we can condition on the 16 binary evidence literals $\neg\, \mathrm{p}(a, a) \wedge \neg\, \mathrm{p}(a, b) \wedge \cdots \wedge \neg\, \mathrm{p}(d, c) \wedge \mathrm{p}(d, d)$ of $\mathbf{P}$ by conditioning on the the 8 unary literals $\neg\, \mathrm{q}(a) \wedge \mathrm{q}(b) \wedge \neg\, \mathrm{q}(c) \wedge \mathrm{q}(d) \wedge \mathrm{r}(a) \wedge \neg\, \mathrm{r}(b) \wedge \neg\, \mathrm{r}(c) \wedge \mathrm{r}(d)$ represented by $\mathbf{q}$ and $\mathbf{r}$.

## 2.3 Matrix-Product Binary Evidence

This idea of encoding a binary relation in unary relations can be generalized to $n$ pairs of unary relations, by adding the following formula to our model.

$$\forall X,\ \forall Y,\ \mathrm{p}(X, Y) \Leftrightarrow (\mathrm{q}_1(X) \wedge \mathrm{r}_1(Y)) \vee (\mathrm{q}_2(X) \wedge \mathrm{r}_2(Y)) \vee \cdots \vee (\mathrm{q}_n(X) \wedge \mathrm{r}_n(Y)) \tag{2}$$

By conditioning on the $\mathrm{q}_i$ and $\mathrm{r}_i$ relations, we can now condition on a much richer set of binary p relations. The relations that can be expressed this way are all the matrices that can be represented by the sum of outer products (in Boolean algebra, where $+$ is $\vee$ and $1 \vee 1 = 1$):

$$\mathbf{P} = \mathbf{q}_1\,\mathbf{r}_1^{\mathsf{T}} \vee \mathbf{q}_2\,\mathbf{r}_2^{\mathsf{T}} \vee \cdots \vee \mathbf{q}_n\,\mathbf{r}_n^{\mathsf{T}} = \mathbf{Q}\,\mathbf{R}^{\mathsf{T}} \tag{3}$$

where the columns of $\mathbf{Q}$ and $\mathbf{R}$ are the $\mathbf{q}_i$ and $\mathbf{r}_i$ vectors respectively, and the matrix multiplication is performed in Boolean algebra, that is,

$$(\mathbf{Q}\,\mathbf{R}^{\mathsf{T}})_{i,j} = \bigvee{}_r \mathbf{Q}_{i,r} \wedge \mathbf{R}_{j,r}$$

**Example 4.** Consider the following $\mathbf{P}$, its decomposition into a sum/disjunction of outer vector products, and the corresponding Boolean matrix multiplication.

$$\mathbf{P} = \begin{bmatrix} 1 & 1 & 0 & 0 \\ 1 & 1 & 0 & 1 \\ 0 & 0 & 1 & 0 \\ 1 & 0 & 0 & 1 \end{bmatrix} = \begin{bmatrix} 0 \\ 1 \\ 0 \\ 1 \end{bmatrix} \begin{bmatrix} 1 \\ 0 \\ 0 \\ 1 \end{bmatrix}^{\mathsf{T}} \vee \begin{bmatrix} 1 \\ 1 \\ 0 \\ 0 \end{bmatrix} \begin{bmatrix} 1 \\ 1 \\ 0 \\ 0 \end{bmatrix}^{\mathsf{T}} \vee \begin{bmatrix} 0 \\ 0 \\ 1 \\ 0 \end{bmatrix} \begin{bmatrix} 0 \\ 0 \\ 1 \\ 0 \end{bmatrix}^{\mathsf{T}} = \begin{bmatrix} 0 & 1 & 0 \\ 1 & 1 & 0 \\ 0 & 0 & 1 \\ 1 & 0 & 0 \end{bmatrix} \begin{bmatrix} 1 & 1 & 0 \\ 0 & 1 & 0 \\ 0 & 0 & 1 \\ 1 & 0 & 0 \end{bmatrix}^{\mathsf{T}}$$

This factorization shows that we can condition on the binary evidence literals of $\mathbf{P}$ (see Example 2) by conditioning on the unary literals

$$\begin{aligned} e = &\ [\neg\, \mathrm{q}_1(a) \wedge \mathrm{q}_1(b) \wedge \neg\, \mathrm{q}_1(c) \wedge \mathrm{q}_1(d)] \wedge [\mathrm{r}_1(a) \wedge \neg\, \mathrm{r}_1(b) \wedge \neg\, \mathrm{r}_1(c) \wedge \mathrm{r}_1(d)] \\ &\wedge [\mathrm{q}_2(a) \wedge \mathrm{q}_2(b) \wedge \neg\, \mathrm{q}_2(c) \wedge \neg\, \mathrm{q}_2(d)] \wedge [\mathrm{r}_2(a) \wedge \mathrm{r}_2(b) \wedge \neg\, \mathrm{r}_2(c) \wedge \neg\, \mathrm{r}_2(d)] \\ &\wedge [\neg\, \mathrm{q}_3(a) \wedge \neg\, \mathrm{q}_3(b) \wedge \mathrm{q}_3(c) \wedge \neg\, \mathrm{q}_3(d)] \wedge [\neg\, \mathrm{r}_3(a) \wedge \neg\, \mathrm{r}_3(b) \wedge \mathrm{r}_3(c) \wedge \neg\, \mathrm{r}_3(d)]. \end{aligned}$$

# 3 Boolean Matrix Factorization

Matrix factorization (or decomposition) is a popular linear algebra tool. Some well-known instances are *singular value decomposition* and *non-negative matrix factorization (NMF)* [17, 18]. NMF factorizes into a product of non-negative matrices, which are more easily interpretable, and therefore attracted much attention for unsupervised learning and feature extraction. These factorizations all work with real-valued matrices. We instead consider Boolean-valued matrices, with only 0/1 entries.

## 3.1 Boolean Rank

Factorizing a matrix $\mathbf{P}$ as $\mathbf{Q}\,\mathbf{R}^{\mathsf{T}}$ in Boolean algebra is a known problem called Boolean Matrix Factorization (BMF) [19, 20]. BMF factorizes a $(k \times l)$ matrix $\mathbf{P}$ into a $(k \times n)$ matrix $\mathbf{Q}$ and a $(l \times n)$ matrix $\mathbf{R}$, where potentially $n \ll k$ and $n \ll l$ and we always have that $n \leq \min(k, l)$.

Any Boolean matrix can be factorized this way and the smallest number $n$ for which it is possible is called the *Boolean rank* of the matrix. Unlike (textbook) real-valued rank, computing the Boolean rank is NP-hard and cannot be approximated unless P=NP [19]. The Boolean and real-valued rank are incomparable, and the Boolean rank can be exponentially smaller than the real-valued rank.

**Example 5.** The factorization in Example 4 is a BMF with Boolean rank 3. It is only a decomposition in Boolean algebra and not over the real numbers. Indeed, the matrix product over the reals contains an incorrect value of 2:

$$\begin{bmatrix} 0 & 1 & 0 \\ 1 & 1 & 0 \\ 0 & 0 & 1 \\ 1 & 0 & 0 \end{bmatrix} \times_{real} \begin{bmatrix} 1 & 1 & 0 \\ 0 & 1 & 0 \\ 0 & 0 & 1 \\ 1 & 0 & 0 \end{bmatrix}^{\mathsf{T}} = \begin{bmatrix} 1 & 1 & 0 & 0 \\ \mathbf{2} & 1 & 0 & 1 \\ 0 & 0 & 1 & 0 \\ 1 & 0 & 0 & 1 \end{bmatrix} \neq \mathbf{P}$$

Note that $\mathbf{P}$ is of full real-valued rank (having four non-zero singular values) and that its Boolean rank is lower than its real-valued rank.

## 3.2 Approximate Boolean Factorization

Computing Boolean ranks is a theoretical problem. Because most real-world matrices will have nearly full rank (i.e., almost $\min(k, l)$), applications of BMF look at approximate factorizations. The goal is to find a pair of (small) Boolean matrices $\mathbf{Q}_{k \times n}$ and $\mathbf{R}_{l \times n}$ such that $\mathbf{P}_{k \times l} \approx \left( \mathbf{Q}_{k \times n}\,\mathbf{R}_{l \times n}^{\mathsf{T}} \right)$, or more specifically, to find matrices that optimize some objective that trades off approximation error and Boolean rank $n$. When $n \ll k$ and $n \ll l$, this approximation extracts interesting structure and removes noise from the matrix. This has caused BMF to receive considerable attention in the data mining community recently, as a tool for analyzing high-dimensional data. It is used to find important and interpretable (i.e., Boolean) concepts in a data matrix.

Unfortunately, the approximate BMF optimization problem is NP-hard as well, and inapproximable [20]. However, several algorithms have been proposed that work well in practice. Algorithms exist that find good approximations for fixed values of $n$ [20], or when $\mathbf{P}$ is sparse [21]. BMF is related to other data mining tasks, such as biclustering [22] and tiling databases [23], whose algorithms could also be used for approximate BMF. In the context of social network analysis, BMF is related to stochastic block models [24] and their extensions, such as infinite relational models.

# 4 Complexity of Binary Evidence

Our goal in this section is to provide a new complexity result for reasoning with binary evidence in the context of lifted inference. Our result can be thought of as a parametrized complexity result, similar to ones based on treewidth in the case of propositional inference. To state the new result, however, we must first define formally the computational task. We will also review the key complexity result that is known about this computation now (i.e., the one we will be improving on).

Consider an MLN $\Delta$ and let $\Gamma_m$ contain a set of ground literals representing binary evidence. That is, for some binary predicate $\mathrm{p}(X, Y)$, evidence $\Gamma_m$ contains precisely one literal (positive or negative) for each grounding of predicate $\mathrm{p}(X, Y)$. Here, $m$ represents the number of objects that parameters $X$ and $Y$ may take.[2] Therefore, evidence $\Gamma_m$ must contain precisely $m^2$ literals.

Suppose now that $\Pr_m$ is the distribution induced by MLN $\Delta$ over $m$ objects, and $q$ is a ground literal. Our analysis will apply to classes of models $\Delta$ that are *domain-liftable* [4], which means that the complexity of computing $\Pr_m(q)$ without evidence is polynomial in $m$. One such class is the set of MLNs with two logical variables per formula [5].

Our task is then to compute the posterior probability $\Pr_m(q|e_m)$, where $e_m$ is a conjunction of the ground literals in binary evidence $\Gamma_m$. Moreover, our goal here is to characterize the complexity of this computation as a function of evidence size $m$.

The following recent result provides a lower bound on the complexity of this computation [11].

**Theorem 1.** *Suppose that evidence $\Gamma_m$ is binary. Then there exists a domain-liftable MLN $\Delta$ with a corresponding distribution $\Pr_m$, and a posterior marginal $\Pr_m(q|e_m)$ that cannot be computed by any algorithm whose complexity grows polynomially in evidence size $m$, unless $P = NP$.*

This is an analogue to results according to which, for example, the complexity of computing posterior probabilities in propositional graphical models is exponential in the worst case. Yet, for these models, the complexity of inference can be parametrized, allowing one to bound the complexity of inference on some models. Perhaps the best example of such a parametrized complexity is the one based on treewidth, which can be thought of as a measure of the model's sparsity (or tree-likeness). In this case, inference can be shown to be linear in the size of the model and exponential only in its treewidth. Hence, this parametrized complexity result allows us to state that inference can be done efficiently on models with bounded treewidth.

We now provide a similar parameterized complexity result, but for evidence in lifted inference. In this case, the parameter we use to characterize complexity is that of Boolean rank.

**Theorem 2.** *Suppose that evidence $\Gamma_m$ is binary and has a bounded Boolean rank. Then for every domain-liftable MLN $\Delta$ and corresponding distribution $\Pr_m$, the complexity of computing posterior marginal $\Pr_m(q|e_m)$ grows polynomially in evidence size $m$.*

The proof of this theorem is based on the reduction from binary to unary evidence, which is described in Section 2. In particular, our reduction first extends the MLN $\Delta$ with Formula 2, leading to the new MLN $\Delta'$ and new pairs of unary predicates $q_i$ and $r_i$. This does not change the domain-liftability of $\Delta'$, as Formula 2 is itself liftable. We then replace binary evidence $\Gamma_m$ by unary evidence $\Gamma'$. That is, the ground literals of the binary predicate p are replaced by ground literals of the unary predicates $q_i$ and $r_i$ (see Example 4). This unary evidence is obtained by Boolean matrix factorization. As the matrix size in our reduction is $m^2$, the following Lemma implies that the first step of our reduction is polynomial in $m$ for bounded rank evidence.

**Lemma 3** (Miettinen [25]). *The complexity of Boolean matrix factorization for matrices with bounded Boolean rank is polynomial in their size.*

The main observation in our reduction is that Formula 2 has size $n$, which is the Boolean rank of the given binary evidence. Hence, when the Boolean rank $n$ is bounded by a constant, the size of the extended MLN $\Delta'$ is independent of the evidence size and is proportional to the size of the original MLN $\Delta$.

We have now reduced inference on MLN $\Delta$ and binary evidence $\Gamma_m$ into inference on an extended MLN $\Delta'$ and unary evidence $\Gamma'$. The second observation behind the proof is the following.

**Lemma 4** (Van den Broeck and Davis [11], Van den Broeck [26]). *Suppose that evidence $\Gamma_m$ is unary. Then for every domain-liftable MLN $\Delta$ and corresponding distribution $\Pr_m$, the complexity of computing posterior marginal $\Pr_m(q|e_m)$ grows polynomially in evidence size $m$.*

Hence, computing posterior probabilities can be done in time which is polynomial in the size of unary evidence $m$, which completes our proof.

We can now identify additional similarities between treewidth and Boolean rank. Exact inference algorithms for probabilistic graphical models typically perform two steps, namely to **(a)** compute a tree decomposition of the graphical model (or a corresponding variable order), and **(b)** perform inference that is polynomial in the size of the decomposition, but potentially exponential in its (tree)width. The analogous steps for conditioning are to **(a)** perform a BMF, and **(b)** perform inference that is polynomial in the size of the BMF, but potentially exponential in its rank. The **(a)** steps are both NP-hard, yet are efficient assuming bounded treewidth [27] or bounded Boolean rank (Lemma 3). Whereas

treewidth is a measure of tree-likeness and sparsity of the graphical model, Boolean rank seems to be a fundamentally different property, more related to the presence of symmetries in evidence.

## 5  Over-Symmetric Evidence Approximation

Theorem 2 opens up many new possibilities. Even for evidence with high Boolean rank, it is possible to find a *low-rank approximate BMF* of the evidence, as is commonly done for other data mining and machine learning problems. Algorithms already exist for solving this task (cf. Section 3).

**Example 6.**  The evidence matrix from Example 4 has Boolean rank three. Dropping the third pair of vectors reduces the Boolean rank to two.

$$
\begin{bmatrix} 1 & 1 & 0 & 0 \\ 1 & 1 & 0 & 1 \\ 0 & 0 & 1 & 0 \\ 1 & 0 & 0 & 1 \end{bmatrix} \approx \begin{bmatrix} 0 \\ 1 \\ 0 \\ 1 \end{bmatrix}\begin{bmatrix} 1 \\ 0 \\ 0 \\ 1 \end{bmatrix}^{\mathsf{T}} \vee \begin{bmatrix} 1 \\ 1 \\ 0 \\ 0 \end{bmatrix}\begin{bmatrix} 1 \\ 1 \\ 0 \\ 0 \end{bmatrix}^{\mathsf{T}} \vee \cancel{\begin{bmatrix} 0 \\ 0 \\ 1 \\ 0 \end{bmatrix}\begin{bmatrix} 0 \\ 0 \\ 1 \\ 0 \end{bmatrix}^{\mathsf{T}}} = \begin{bmatrix} 0 & 1 \\ 1 & 1 \\ 0 & 0 \\ 1 & 0 \end{bmatrix}\begin{bmatrix} 1 & 1 \\ 0 & 1 \\ 0 & 0 \\ 1 & 0 \end{bmatrix}^{\mathsf{T}} = \begin{bmatrix} 1 & 1 & 0 & 0 \\ 1 & 1 & 0 & 1 \\ 0 & 0 & \mathbf{0} & 0 \\ 1 & 0 & 0 & 1 \end{bmatrix}
$$

This factorization is approximate, as it flips the evidence for atom $\mathrm{p}(c, c)$ from true to false (represented by the bold 0). By paying this price, the evidence has more symmetries, and we can condition on the binary relation by introducing only two instead of three new pairs $(\mathrm{q}_i, \mathrm{r}_i)$ of unary predicates.

Low-rank approximate BMF is an instance of a more general idea; that of *over-symmetric evidence approximation*. This means that when we want to compute $\Pr(q \mid e)$, we approximate it by computing $\Pr(q \mid e')$ instead, with evidence $e'$ that permits more efficient inference. In this case, it is more efficient because it maintains more symmetries of the model and permits more lifting. Because all lifted inference algorithms, exact or approximate, exploit symmetries, we expect this general idea, and low-rank approximate BMF in particular, to improve the performance of any lifted inference algorithm.

Having a small amount of incorrect evidence in the approximation need not be a problem. As these literals are not covered by the first most important vector pairs, they can be considered as noise in the original matrix. Hence, a low-rank approximation may actually improve the performance of, for example, a lifted collective classification algorithm. On the other hand, the approximation made in Example 6 may not be desirable if we are querying attributes of the constant $c$, and we may prefer to approximate other areas of the evidence matrix instead. There are many challenges in finding appropriate evidence approximations, which makes the task all the more interesting.

## 6  Empirical Evaluation

To complement the theoretical analysis from the previous sections, we will now report on experiments that investigate the following practical questions.

**Q1**  How well can we approximate a real-world relational data set by a low-rank Boolean matrix?

**Q2**  Is Boolean rank a good indicator of the complexity of inference, as suggested by Theorem 2?

**Q3**  Is over-symmetric evidence approximation a viable technique for approximate lifted inference?

To answer **Q1**, we compute approximations of the linkto binary relation in the WebKB data set using the ASSO algorithm for approximate BMF [20]. The WebKB data set consists of web pages from the computer science departments of four universities [28]. The data has information about words that appear on pages, labels of pages and links between web pages (linkto relation). There are four folds, one for each university. The exact evidence matrix for the linkto relation ranges in size from 861 by 861 to 1240 by 1240. Its real-valued rank ranges from 384 to 503. Performing a BMF approximation in this domain adds or removes hyperlinks between web pages, so that more web pages can be grouped together that behave similarly.

Figure 1 plots the approximation error for increasing Boolean ranks, measured as the number of incorrect evidence literals. The error goes down quickly for low rank, and is reduced by half after Boolean rank 70 to 80, even though the matrix dimensions and real-valued rank are much higher. Note that these evidence matrices contain around a million entries, and are sparse. Hence, these approximations correctly label $99.7\%$ to $99.95\%$ of the atoms.

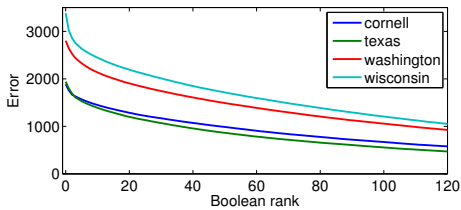

| Rank $n$ | Circuit Size (a) | Circuit Size (b) |
|:---:|:---:|:---:|
| 0 | 18 | 24 |
| 1 | 58 | 50 |
| 2 | 160 | 129 |
| 3 | 1873 | 371 |
| 4 | $> 2129$ | 1098 |
| 5 | ? | 3191 |
| 6 | ? | 9571 |

Figure 1: Approximation BMF error in terms of the number of incorrect literals for the WebKB linkto relation.

Figure 2: First-order NNF circuit size (number of nodes) for increasing Boolean rank $n$, and (a) the peer to peer and (b) hierarchical model.

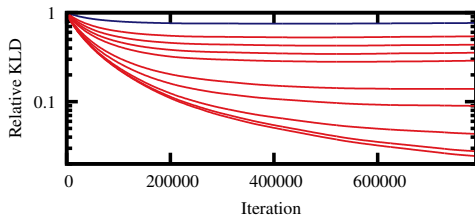

(a) Texas Data Set

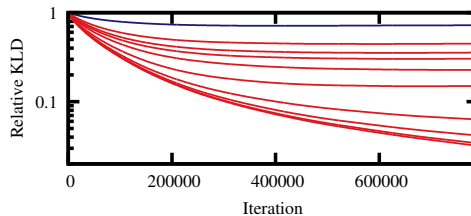

(b) Wisconsin Data Set

Figure 3: KLD of LMCMC on different BMF approximations, relative to the KLD of vanilla MCMC on the same approximation. From top to bottom, the lines represent exact evidence (blue), and approximations (red) of rank 150, 100, 75, 50, 20, 10, 5, 2, and 1.

To answer **Q2**, we perform two sets of experiments. Firstly, we look at *exact* lifted inference and investigate the influence of adding Formula 2 to the "peer-to-peer" and "hierarchical" MLNs from Example 1. The goals is to condition on linkto relations with increasing rank $n$. These models are compiled using the WFOMC [8] algorithm into first-order NNF circuits, which allow for exact domain-lifted inference (c.f., Lemma 4). Table 2 shows the sizes of these circuits. As expected, circuit sizes grow exponentially with $n$. Evidence breaks more symmetries in the peer-to-peer model than in the hierarchical model, causing the circuit size to increase more quickly with Boolean rank.

Since the connection between rank and exact inference is obvious from Theorem 2, the more interesting question in **Q2** is whether Boolean rank is indicative of the complexity of *approximate* lifted inference as well. Therefore, we investigate its influence on the Lifted MCMC algorithm (LMCMC) [29] with Rao-Blackwellized probability estimation [30]. LMCMC interleaves standard MCMC steps (here Gibbs sampling) with jumps to states that are symmetric in the graphical model, in order to speed up mixing of the chain. We run LMCMC on the WebKB MLN of Davis and Domingos [31], which has 333 first-order formulas and over 1 million random variables. It classifies web pages into 6 categories, based on their link structure and the 50 most predictive words they contain. We learn its parameters with the Alchemy package and obtain evidence sets of varying Boolean rank from the factorizations of Figure 1.[3] For these, we run both vanilla and lifted MCMC, and measure the KL divergence (KLD) between the marginal distribution at each iteration[4], and a ground truth obtained from 3 million iterations on the corresponding evidence set. Figure 3 plots the KLD of LMCMC divided by the KLD of MCMC. It shows that the improvement of LMCMC over MCMC goes down with Boolean rank, answering **Q2** positively.

To answer **Q3**, we look at the KLD between different evidence approximations $\Pr(.\,|\,e'_n)$ of rank $n$, and the true marginals $\Pr(.\,|\,e)$ conditioned on exact evidence. As this requires a good estimate of $\Pr(.\,|\,e)$, we make our learned WebKB model more tractable by removing formulas about word content. For two approximations $e'_a$ and $e'_b$ such that rank $a < b$, we expect LMCMC to converge faster to $\Pr(.\,|\,e'_a)$ than to $\Pr(.\,|\,e'_b)$, as suggested by Figure 3. However, because $\Pr(.\,|\,e'_a)$ is a more crude approximation of $\Pr(.\,|\,e)$ than $\Pr(.\,|\,e'_b)$ is, the KLD at convergence should be worse for $a$

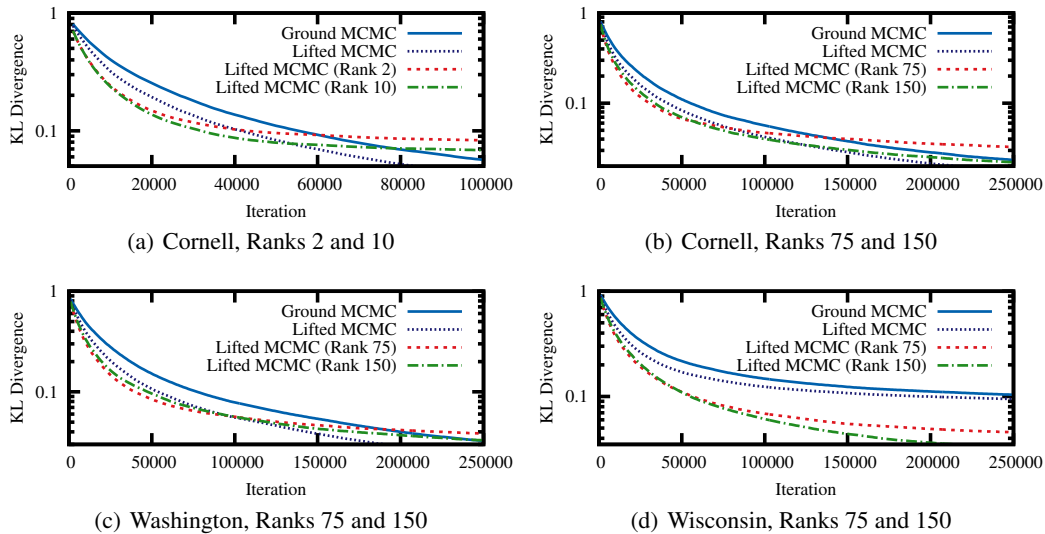

Figure 4: Error for different low-rank approximations of WebKB, in KLD from true marginals.

than for $b$. Hence, we expect to see a *trade-off*, where the lowest ranks are optimal in the beginning, higher ranks become optimal later one, and the exact model is optimal at convergence.

Figure 4 shows exactly that, for a representative sample of ranks and data sets. In Figure 4(a), rank 2 and 10 outperform LMCMC with the exact evidence at first. Exact evidence overtakes rank 2 after 40k iterations, and rank 10 after 50k. After 80k iterations, even non-lifted MCMC outperforms these crude approximations. Figure 4(b) shows the other side of the spectrum, where a rank 75 and 150 approximation are overtaken at iterations 90k and 125k. Figure 4(c) is representative of other datasets. Note here that at around iteration 50k, rank 75 in turn outperforms the rank 150 approximation, which has fewer symmetries and does not permit as much lifting. Finally, Figure 4(d) shows the ideal case for low-rank approximation. This is the largest dataset, and therefore the most challenging inference task. Here, LMCMC on $e$ converges slowly compared to its approximations $e'$, and $e'$ results in almost perfect marginals. The crossover point where exact inference outperforms the approximation is never reached in practice. This answers **Q3** positively.

# 7 Conclusions

We presented two main results. The first is a more precise complexity characterization of conditioning on binary evidence, in terms of its Boolean rank. The second is a technique to approximate binary evidence by a low-rank Boolean matrix factorization. This is a first type of over-symmetric evidence approximation that can speed up lifted inference. We showed empirically that low-rank BMF speeds up approximate inference, leading to improved approximations.

For future work, we want to evaluate the practical implications of the theory developed for other lifted inference algorithms, such as lifted BP, and look at the performance of over-symmetric evidence approximation on machine learning tasks such as collective classification. There are many remaining challenges in finding good evidence-approximation schemes, including ones that are query-specific (cf. de Salvo Braz et al. [32]) or that incrementally run inference to find better approximations (cf. Kersting et al. [33]). Furthermore, we want to investigate other subsets of binary relations for which conditioning could be efficient, in particular functional relations $p(X, Y)$, where each $X$ has at most a limited number of associated $Y$ values.

### Acknowledgments

We thank Pauli Miettinen, Mathias Niepert, and Jilles Vreeken for helpful suggestions. This work was supported by ONR grant #N00014-12-1-0423, NSF grant #IIS-1118122, NSF grant #IIS-0916161, and the Research Foundation-Flanders (FWO-Vlaanderen).

## Footnotes

[1]Partial evidence on the relation $\mathrm{p}$ can be encoded as full evidence on predicates $\mathrm{p}_0$ and $\mathrm{p}_1$ by adding formulas $\forall X, Y \ \mathrm{p}(X, Y) \Leftarrow \mathrm{p}_1(X, Y)$ and $\forall X, Y \ \neg \mathrm{p}(X, Y) \Leftarrow \mathrm{p}_0(X, Y)$ to the model.

[2] We assume without loss of generality that all logical variables range over the same set of objects.

[3] When synthetically generating evidence of these ranks, results are comparable.

[4] Runtime per iteration is comparable for both algorithms. BMF runtime is negligible.

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
