[Reviews · NeurIPS 2013]

Submitted by Assigned_Reviewer_1

Every so often, you read a paper that makes you think, "Wow, I wish I'd thought of that!" This is one such paper.

This paper presents a method to do efficient lifted inference, even when there is evidence on binary predicates. This is a very significant contribution because lifted inference algorithms tend to scale very poorly with the amount of evidence available. It was recently shown that evidence on unary predicates can be efficiently incorporated into exact lifted inference algorithms, but that evidence on binary predicates, in general, cannot. This paper describes the special cases when such evidence can be incorporated efficiently: when the predicate's evidence matrix has low Boolean rank, it can be transformed into a set of unary predicates that can be efficiently incorporated as evience. When the evidence is not low-rank (which is the more common case), it may still be worth approximating it as a low-rank matrix in order to reap the benefits of lifted inference. This paper also shows how to do this and presents some basic experimental results in one domain.

This paper is very clearly written and contains a very significant theoretical concept that is key to lifted inference. At its core, the idea is very simple, but I predict that it will have a large impact in the lifted inference community. Most existing lifted inference algorithms are not much better than ground inference algorithms when there is a lot of evidence. This paper brings us one step closer to making lifted inference much more practical.

As far as weaknesses, the experimental results suggest that although the low-rank approximation leads to faster lifted inference, it also hurts accuracy. After enough iterations, ground MCMC is usually more accurate than lifted MCMC, even when the approximation has a relatively large rank (such as 75 or 150). Exact inference is constrained to much smaller rank approximations (< 10). It would be very interesting to see more experimental results on other datasets and using other algorithms, such as lifted BP, and it would be nice to see more realistic cases where the low-rank approximation leads to much better results.

Even so, the experiments do a good job of clearly demonstrating that the low-rank approximation leads to faster convergence of lifted MCMC due to the additional symmetries, as well as exploring the tradeoffs. Thus, I do not consider these weaknesses to be critical -- expanding the experiments can certainly be left for future work or a journal version of this paper.
Summary: This paper contains a very significant theoretical result about the tractability of lifted inference that is likely to be high-impact. The experiments are preliminary, but do a good job of showing the tradeoffs of applying the proposed approximation on an interesting dataset.

Submitted by Assigned_Reviewer_4

This paper proposes an approach to performing lifted inference in models for statistical relational learning when evidence is binary, i.e., relations between entities, and complex, i.e., symmetry breaking. Leveraging symmetry has been crucial to the success of previous work on lifted inference, but, the authors point out, many real-world problems do not have a great deal of symmetry to leverage. The authors address this problem by first introducing a theoretical description of the complexity of binary evidence, called Boolean rank, and relate that to the complexity of performing inference using that evidence. They then introduce a scheme for approximating evidence of high Boolean rank with similar evidence of a lower Boolean rank and using this approximate evidence for lifted inference. They evaluate their approach on WebKB data and compare lifted and non-lifted inference using the original evidence and evidence approximations of varying Boolean rank. By approximating evidence, they are able to improve speed without sacrificing a great deal of predictive accuracy.

This is a very well written paper. The exposition is clear, the problem investigated is an important one, and the introduced theory and strategy are original and compelling solutions.

A weakness of the paper is that only WebKB is considered. It would be nice to see if the paper's approach could scale Markov logic networks to bigger problems than WebKB.

One style note: NIPS style is to number the introduction as section 1.
Summary: This paper introduces new theory and strategy for increasing the applicability of lifted inference by approximating evidence with modified evidence more conducive to lifted inference. It is a new approach to an important problem.

Submitted by Assigned_Reviewer_6

I should be upfront that though I am quite familiar with graphical models and Markov Random Fields, I'm far from an expert on Markov Logic Networks (MLNs). So I approached this paper through the lens of a generic graphical models person, and inevitably I tended to interpret the results in that way.

This paper is interested in inference (computing marginal probabilities and/or a normalization factor) in a MLN. Inference is "conditional" in the sense that one will first observe some "evidence", which gives a subset of allowed values to each variable or pair of variables. Evidence is "unary" if it constrains single variables only, and "binary" if it also constraints pairs. The main idea is the following: Though inference in large MLNs is generally hard, some existing "lifted" algorithms can perform inference efficiently in large models by exploiting special structure in special classes of MLNs. However, conditioning on binary evidence destroys this special structure.

This paper basically points out that, rather than conditioning on binary evidence, one could augment the graph with extra variables, and then condition on only on unary evidence. However, in the worst case, the needed number of new variables is exponential in the number of values each variable can take. However, if the binary evidence for a given pair happens to be (binary) low-rank, then a much smaller number of variables are needed. Thus, inference in general can be approached by (possibly approximately) finding a low-rank decomposition of each binary evidence term, adding extra variables, and then using the specialized unary inference algorithm.

The major technical result is Theorem 2 which states that if you have a "lifted" inference algorithm and all pairwise evidence admits a low-rank decomposition, than exact inference can be done in polynomial time. This can naturally (i.e. heuristically) be done approximately through approximate decomposition.

A couple other issues:

- Is Boolean rank essential? Could one alternatively state that inference is always exponential in the number of values that each variable can have? (Which is weaker than Boolean rank, but simpler to understand.) The experiments use a very large number of variables, which motivates the decomposition, but presumably many applications have small numbers, in which case decomposition is superfluous. Is this result still useful?

- Theorem 1 is strangely stated. I think this is previous work, in which case a reference needs to be provided. I also think that one should say "no polynomial time algorithm exists" rather than "takes exponential time".

- Finally, for what its worth, as someone quite familiar with graphical models and MRFs, but not so familiar with Markov logic networks, I found this paper quite challenging to understand. The major issue is that very many technical terms ("evidence", "term", "literal", "relation", "inference", etc.) are used undefined, and there is no clear reference provided where these terms are precisely defined. After reading quite a few of the background papers I think I understand these, but the literature on MLNs doesn't seem totally consistent in usage, which makes things a challenge. (For example, typically "evidence" is a set of fixed values for a set of variables, whereas this paper uses it as a set of constraints on variables or pairs of variables.) At a minimum, the paper should state what is meant by inference, summarize what classes of symmetries previous work has taken advantage of to make inference tractable, and be precise about what evidence is and how it changes inference. It could be than in an eight page paper there is no way to make the current paper easily understandable by readers that aren't MLN experts, but I do think more effort could be made in this direction.


Edit after author response:

In my initial review, I had a concern about how conditioning could affect the efficiency of inference. After reading the other reviews, the author response, and discussing with the other reviewers, I'm convinced this isn't an issue, and am correspondingly raising my rating.
Summary: The idea seems OK, though a moderate increment on existing results. I would lean towards being more positive if I understood why the main result is true, which I hope can be addressed in discussions.

Submitted by Assigned_Reviewer_7

This paper explores how Boolean rank can be be used to characterize the efficiency of lifted inference for binary evidence (but, interestingly not beyond binary).

Unfortunately there are some very misleading statements in the paper which would lead the reader to make some erroneous conclusions. For example "They perform inference that is polynomial in the number of objects in the model [5] and are therein exponentially faster than classical inference algorithms." is only true for very restricted conditions (what the authors later call "domain-liftable"). [5] only proves it for formulae that only contain up-to two logical variables. Moreover it isn't true in general. You need to be much more precise in your claims.

You need to give a reference for theorem 1. It isn't new to this paper.
Summary: This is a good paper that explores how, even though how lifted inference with observations of more than 2 arguments is #P-hard, it can be efficient if the observations have a low Boolean rank. This observation leads to an approximation scheme to find an approximation with low Boolean rank.
Author Feedback

Author rebuttal: We thank the reviewers for their kind remarks.

In response to Reviewer 4, on the use of WebKB, we want to point out that WebKB with binary evidence is a far harder problem than what is typically considered in lifted inference papers (even most approximate ones). It has >1 million random variables and factors, and 333 MLN formulas.
We see the use of WebKB as a strength of the paper.

We do agree with Reviewers 1 and 4 that more experiments, with more data sets and algorithms, would be very interesting. Given the space constraints, we believe these should be left for future work. We also hope that other lifted inference authors will take the general ideas presented here and apply them to their specific algorithms.

In response to Reviewer 6, we understand that the paper can be challenging for people unfamiliar with statistical relational learning. We want to address this based on your feedback, in so far that space allows us.


We would like to clarify the following for Reviewer 6 (based on the initial review):

- There seems to be a mix-up between random variables (ground atoms) and logical variables in your review.
First-order atoms with constraints on the logical variables represent sets of random variables. Similarly, first-order formulas represent sets of factors. These are used by lifted inference algorithms to reason about groups of random variables or factors efficiently, without grounding.
Evidence is a truth-value assignment to a set of random variables (as in Bayesian networks), which can equivalently be represented by a truth-value assignment to first-order atoms with constraints on the logical variables.

- Viewing our work as "augmenting the graph with additional random variables and conditioning on them" is a bad premise to understand the contributions of the paper.
Lifted inference is all about the symmetries expressed by a first-order formulation of the model. A key insight of lifted inference is that the complexity of inference can be exponential in the size of this first-order formulation, but only polynomial in the domain size, that is, the number of objects in the world. When looking only at the induced probabilistic graphical model (PGM), these two parameters blend into one, namely the size of the PGM, and those insights are lost.
This paper is all about distinguishing two parameters for evidence: the size of the evidence (~domain size) and its Boolean rank (~size of first-order representation). Our key insight is that our encoding increases the size of the first-order representation with the Boolean rank, and not with the size of the evidence. Hence the complexity of inference will be polynomial if the Boolean rank is bounded.
As you can see, distinguishing these two parameters is essential in understanding our work, and the difference is lost when looking at the PGM.

- About the suggestion to "state that inference is exponential in the number of values variables can have": this is not the case for lifted inference (assuming you mean logical variables). In fact, the definition of domain-lifted inference states exactly the opposite [4].

- About the comment that our experiments use a very large number of variables, and many applications have a small number: this is not the case in statistical relational learning. MLNs are typically small, but they are applied to very large graphs, such as social networks, or web pages, which lead to very large domain sizes, and a very large number of random variables.

- About the changed structure of \Delta' being a problem: the structure of Formula 2 has no influence on the 'liftability' of the model \Delta'. It beaks no symmetries, and it is easily handled by modern lifted inference algorithms. For example, lifted inference algorithms FOVE, PTP and WFOMC can all eliminate unary atoms, reducing Formula 2 to a unit clause, which can be propagated or absorbed. In short, it poses no problem. We will say that more explicitly.

- The mapping from binary to unary evidence does not require an exponential number of new variables (random or logical). The number of additional unary predicates, first-order atoms, and logical variables added to the model is all linear in the Boolean rank and independent of the domain or evidence size. The number of additional random variables is polynomial in the domain or evidence size.
The exponential blowup comes from the fact that Formula 2 has size linear in the Boolean rank. Because in general, domain-lifted inference can be exponential in the size of the MLN (not the domain size), adding Formula 2 to \Delta' makes that inference can be exponential in Boolean rank, but not the size of the evidence.

- The reference for Theorem 1 is [11]. We agree with your suggestion there.